# Estimating Overall and Cause-Specific Excess Mortality during the COVID-19 Pandemic: Methodological Approaches Compared

**DOI:** 10.3390/ijerph20115941

**Published:** 2023-05-24

**Authors:** Claudio Barbiellini Amidei, Ugo Fedeli, Nicola Gennaro, Laura Cestari, Elena Schievano, Manuel Zorzi, Paolo Girardi, Veronica Casotto

**Affiliations:** 1Epidemiological Department, Azienda Zero, Veneto Region, 35131 Padova, Italy; 2Department of Environmental Sciences, Informatics and Statistics, Ca’ Foscari University of Venice, 30172 Venice, Italy

**Keywords:** mortality, COVID-19, epidemiological methods, excess mortality, forecasted mortality

## Abstract

During the COVID-19 pandemic, excess mortality has been reported worldwide, but its magnitude has varied depending on methodological differences that hinder between-study comparability. Our aim was to estimate variability attributable to different methods, focusing on specific causes of death with different pre-pandemic trends. Monthly mortality figures observed in 2020 in the Veneto Region (Italy) were compared with those forecasted using: (1) 2018–2019 monthly average number of deaths; (2) 2015–2019 monthly average age-standardized mortality rates; (3) Seasonal Autoregressive Integrated Moving Average (SARIMA) models; (4) Generalized Estimating Equations (GEE) models. We analyzed deaths due to all-causes, circulatory diseases, cancer, and neurologic/mental disorders. Excess all-cause mortality estimates in 2020 across the four approaches were: +17.2% (2018–2019 average number of deaths), +9.5% (five-year average age-standardized rates), +15.2% (SARIMA), and +15.7% (GEE). For circulatory diseases (strong pre-pandemic decreasing trend), estimates were +7.1%, −4.4%, +8.4%, and +7.2%, respectively. Cancer mortality showed no relevant variations (ranging from −1.6% to −0.1%), except for the simple comparison of age-standardized mortality rates (−5.5%). The neurologic/mental disorders (with a pre-pandemic growing trend) estimated excess corresponded to +4.0%/+5.1% based on the first two approaches, while no major change could be detected based on the SARIMA and GEE models (−1.3%/+0.3%). The magnitude of excess mortality varied largely based on the methods applied to forecast mortality figures. The comparison with average age-standardized mortality rates in the previous five years diverged from the other approaches due to the lack of control over pre-existing trends. Differences across other methods were more limited, with GEE models probably representing the most versatile option.

## 1. Introduction

Since 2020, the COVID-19 pandemic has severely affected populations worldwide, increasing mortality for conditions directly related to COVID-19 as well as indirect consequences such as those associated with compromised access to care [1]. Given the complexity of the effects of COVID-19 on mortality, and the possible under-diagnosis of COVID-19 in specific contexts, especially at the beginning of the pandemic, excess all-cause mortality, as opposed to COVID-19-related mortality, has been considered a more appropriate and reliable measure to assess to the impact of the epidemic [2]. Between January 2020 and December 2021, the WHO estimated that there have been up to 14.9 million deaths more than expected [1]. Numerous reports and articles from all over the world have described an unprecedented excess in all-cause mortality [3,4,5,6]. However, the magnitude of this excess varied significantly across studies. Since its appearance, COVID-19 has certainly affected countries differently, but, in addition to this, the use of different methods to estimate excess mortality contributed to increasing the uncertainty on the actual COVID-19 mortality burden [2]. In fact, the estimates of excess mortality strongly depend on the method employed to forecast mortality figures. Differences in forecasted values end up artificially increasing or decreasing the death toll attributed to COVID-19. Several approaches have been adopted, but no single method can be considered preferable to the others. Crucial elements that should be considered in predicting mortality are the presence of preexisting long-term trends and seasonality. However, not all methods could adequately control for these components. In this article, we will compare four different approaches to estimate excess mortality among those most commonly applied in recent literature. These comprise: the use of a simple monthly average of the preceding years as reference (1) [7,8,9]; more sophisticated approaches, based on average-standardized mortality rates (2) [10,11]; models that forecast values based on previous trends and seasonality (Seasonal Autoregressive Integrated Moving Average–SARIMA) (3) [12]; or other models, where further variables can be implemented (Generalized Estimating Equations–GEE models) (4) [13,14]. Given the relevance of correctly quantifying the COVID-19 mortality burden, the importance of accurately assessing the impact of public health policies, and the benefits of mass vaccination on mortality reduction, awareness of the effects of the methodological differences on excess mortality estimates is of crucial importance, along with the promotion of a methodological gold standard.

To ensure homogeneity in the study population, we examined mortality data from the Veneto Region (Northeastern Italy, about 4.9 million inhabitants). The healthcare service in the region is free of charge and funded by general taxation. The region was among the first and most severely hit areas by the pandemic at the beginning of 2020, and this led to a strong public health response [15]. Our aim was to quantify differences in excess death estimates during 2020 by comparing different methodological approaches. We have considered overall mortality, and, to mimic settings with specific mortality trends, we have also analyzed broad nosological categories characterized by both pre-existing decreasing (i.e., circulatory diseases) and increasing trends (i.e., neurologic and mental disorders).

## 2. Materials and Methods

The mortality register of the Veneto Region includes all diseases mentioned in death certificates coded according to the International Classification of Diseases, 10th Revision (ICD-10). The underlying cause of death (UCOD) was selected from all conditions reported on certificates according to rules set by the World Health Organization. To standardize UCOD assignment, the Automated Classification of Medical Entities (ACME) software was applied to regional records until 2017 [16]. From 2018, the IRIS software was adopted, as in most European countries [17]. Deaths of residents in Veneto from 1 January 2008 to 31 December 2020 were included. Analyses were carried out for (1) all-cause mortality and for the following UCOD: (2) neoplasms (C00-D48), (3) circulatory diseases (I00-I99, I46.x excluded), and (4) diseases of the nervous system and mental and behavioral disorders (F00-F99; G00-G99). Both the monthly number of deaths and the monthly age-standardized mortality rates, calculated using the 2013 European as standard population, were reported.

Excess mortality in 2020 was investigated as the percent variation between observed and expected mortality. Four different methodological approaches were used to estimate expected mortality if the pandemic had not occurred (baseline mortality): (1) average monthly number of deaths by cause registered in 2018–2019, (2) average cause specific age-standardized mortality rate in the previous five years (2015–2019), (3) monthly forecasted rates obtained by Seasonal Autoregressive Integrated Moving Average (SARIMA) models, and (4) by Generalized Estimating Equations (GEE) models.

### 2.1. Number of Deaths

The percentage excess was simply the ratio between the monthly number of deaths observed in 2020 and the corresponding average registered in the year 2018 and 2019. 

### 2.2. Age-Standardized Mortality Rate

The percentage excess was the ratio between age-standardized rates observed in 2020 and the mean age-standardized death rate registered in the period 2015–2019.

### 2.3. SARIMA

Using historical mortality trends from 2008 to 2019, SARIMA models were fitted to estimate the expected monthly age-standardized mortality rates in 2020. The seasonal SARIMA model is an extension of the ARIMA (Autoregressive Integrated Moving Average) model that explicitly supports direct modeling of the seasonal component of time-series data. The components included in the SARIMA model were selected by applying an automatic algorithm that minimized the Akaike Information Criterion (AIC). 

### 2.4. GEE

Using historical mortality trends from 2008 to 2019, GEE models were fitted to estimate the expected monthly age-standardized mortality rates in 2020. Considering the presence of only positive values for standardized mortality, we modelled rates assuming a Gamma distribution with log-link function. A first-order autoregressive structure was incorporated in the GEE model to take into account the different pattern of correlation among monthly rates within each year, assuming the independence of the years. Predictors included in the model were dummy variables for each month (with January as the reference category) and a linear trend across months of the study period. The presence of a non–linear trend was tested by adding orthogonal polynomial terms (quadratic and cubic) in the models. However, these terms were statistically non-significant, with the exception of the quadratic term in mortality for neurologic diseases and mental disorders, but no relevant differences in the estimates could be observed. 

All analyses were undertaken using Stata Software 16.0 version. SARIMA models were analyzed by means of the “sarima” package, using R Statistical Software version 4.2.1.3.

## 3. Results

Figure 1 shows monthly age-standardized mortality rate trends in the Veneto Region from January 2008 to December 2020, overall and by the three selected underlying causes of death. A declining long-term trend can be observed for all-cause and circulatory diseases mortality and, to a lesser extent, for mortality from neoplasms. On the other hand, mortality due to neurologic and mental disorders (mainly accounted in Italy by dementia and neurodegenerative diseases) showed an increasing trend. All figures, except for malignant neoplasms, were characterized by a strong seasonal component, with peaks during influenza epidemic months (mainly January or February). In 2020, overall age-standardized mortality rates in January and February were unusually low, while a delayed peak occurred in March–April, associated with the first COVID-19 epidemic wave that hit the region. Very high mortality rates could be observed in November and December, due to the second epidemic wave. In 2020, the three cause-specific mortality rates did not display major changes, except for circulatory diseases showing an abrupt interruption of the pre-existing declining trend. 

In total, 56,973 deaths were recorded in the region in 2020, corresponding to a yearly age-standardized mortality rate of 890.50 per 100,000, as reported in Table 1. Based on the simplest approach to estimate all-cause excess mortality, the comparison of deaths observed in 2020, with the average number in 2018 and 2019, a +17.2% increase was estimated. If monthly age-standardized mortality rates in 2020 were compared to the corresponding average from the previous five years (without taking into account the pre-existing declining trend), the estimated excess amounted to +9.5%. When monthly 2020 rates were forecasted, employing the SARIMA and GEE models, the estimated excess in overall mortality rates through 2020 corresponded to +15.2% and +15.7%, respectively. Estimated excess deaths in the two pandemic peaks (March–April and November–December) were relatively similar considering the simple comparison of death counts, or SARIMA and GEE models for mortality rates, while those obtained using average age-standardized mortality rates were sensibly lower (Figure 2). The relatively low mortality recorded in January 2020 translated in a negative variation estimated with all methods, although especially low values were observed when using the previous 5-year age-standardized average mortality rates (−12.0%) and the SARIMA model (−9.7%), which were more likely influenced by the seasonal component, further affected by two influenza peaks in January 2015 and 2017. Notably, the excess deaths compared to the average numbers in the previous two years, plotted against the number of COVID-19 deaths (both as underlying cause or any mention in death certificates), showed how COVID-19 deaths could explain most of the excess mortality, especially during the second pandemic wave (Appendix A).

Deaths from circulatory diseases in 2020 were 16,484, with a yearly age-standardized mortality rate of 250.85 per 100,000. Estimated excess deaths in 2020 were +7.1% when compared with the 2018–2019 average, similarly to figures estimated with the SARIMA (+8.4%) and GEE (+7.2%) models. On the other hand, the comparison with the average age-standardized mortality rates in the previous five years led to an estimate of an overall mortality rate reduction (−4.4%) in 2020; such a discrepancy could be attributed to the strong pre-pandemic declining trend that could not be taken into account by this latter approach. 

In total, 14,071 deaths due to neoplasms were recorded in 2020, corresponding to a yearly age-standardized mortality rate of 230.19 per 100,000. Most approaches showed substantially unchanged mortalities in 2020 compared to expected values, ranging from −0.1% to −1.6%; only the comparison with age-standardized rates in the previous five years led to estimate a substantial decline in mortality (−5.5%). The absence of any relevant seasonal patterns made estimates of monthly percent variations overall stable. Mortality in the first pandemic wave (March–April 2020) led to estimate a mild increase in deaths attributed to cancer with most methods (mortality in March ranged from +4.6% to +7.0%, except for average age-standardized mortality rate that led to estimate −0.3%), while, during the second wave, and especially in December, all methods showed lower than expected mortalities (ranging from −9.7% to −3.3%). 

In total, 5549 deaths with neurologic and mental disorders as the underlying cause were recorded in 2020, with a yearly mortality rate of 84.60 per 100,000. The pre-pandemic increasing trend that was not accounted for using the simple comparison of death counts and age-standardized rates led to estimate respectively a +4.0% and +5.1% excess mortality in 2020. Forecasted values using the SARIMA and GEE models, on the other hand, showed much less marked variations, −1.3% and +0.3%, respectively. A similar effect could be observed for both pandemic waves. 

Table 2 shows the results of the GEE models applied to the pre-pandemic time-series data of monthly age-standardized mortality rates: a decreasing trend for all-cause, circulatory, and cancer mortality and an increasing trend for neurological and mental disorders could be confirmed. The strong seasonality observed for all nosological categories, with the exception of neoplasms, was highlighted by the significantly lower coefficients for all months when compared to January, taken as reference. Figure 3 shows monthly age-standardized rates registered in 2020 plotted against expected figures obtained through the GEE and SARIMA models. Compared to forecasted figures obtained via the GEE model, a significant increase in all-cause and circulatory disease mortality could be observed in correspondence with the first (March–April) and the second (October–December) epidemic waves involving the Veneto Region in 2020. A similar significant excess mortality could also be observed for figures forecasted using the SARIMA models when considering all-cause mortality and less evidently also for circulatory diseases. 

## 4. Discussion

Despite the importance of estimating excess mortality in pandemic scenarios, such as COVID-19, no standard approach has been adopted in technical reports or published papers. Several methodologies have been proposed [7,18,19], but researchers often end up choosing an approach over another arbitrarily and rarely perform sensitivity analyses to assess whether different variation estimates would have been otherwise achieved had another approach been implemented. These differences are not always negligible, as confirmed by previous studies that addressed this methodological issue [20], and hinder the comparability of the results as well as the magnitude of the estimated impact of the pandemic in terms of excess mortality. The methods we selected for comparison were the most used in recent applications. The length of the data series used for each method was also based on that generally used by researchers using these approaches. Recently published papers on the topic have, in fact, quantified excess deaths by estimating expected mortality in 2020 by using a simple average of the previous years as a reference [7,8,9] or more sophisticated approaches, including average standardized mortality rates [10,11], the SARIMA [12] and GEE models [13,14], as well as other methodologies [21,22,23,24,25,26]. COVID-19 has affected mortality in different aspects and at different times, thus disentangling this effect from the distortion introduced by the application of different methods can be complex [2]. For the purpose, we examined one population over the same study period, and, to account for differences in pre-existing trends, we analyzed causes of death with both decreasing (circulatory diseases) and increasing trends (neurologic and mental disorders). Furthermore, to better examine the effect of different approaches on the seasonal part of trends, we chose an additional cause of death without strong seasonal variation and only a slightly declining long-term trend (neoplasms). To the best of our knowledge, this is the first study to address methodological differences on cause-specific excess mortality during the pandemic. 

Overall, the results from the present study show a strong variability in excess mortality estimates during 2020 based on the cause of death considered and on the different methods applied. For causes with a decreasing pre-pandemic trend (with or without a strong seasonal component), most approaches provided overall similar results, with the exception of age-standardized mortality rates of the previous five years, which led to underestimating the actual impact of the pandemic. When examining causes of death with an increasing trend, analyses based on the average number of deaths and on average age-standardized mortality rates led to estimate a similarly elevated mortality increase, while the variation appeared to be less marked with the SARIMA and, especially, GEE models. This is likely related to the capacity of the latter approaches to intercept the pre-existing increasing trend, as well as the seasonal variations. Preexisting increasing or decreasing trends strongly affect predicted figures. In fact, in the presence of linear trends, the expected month-specific mortality rates are likely to be, respectively, higher or lower than those recorded ever before, something that cannot be achieved by simply using average values from previous years. The distortion is greater, the longer the reference period. When considering long time-series data, the shape of these trends should also be examined, as the change may be non-linear and quadratic, or cubic terms may be more appropriate to describe the trend. 

Given the specificities of the different methods, the period chosen as reference was different. The simplest approach was to compare observed numbers with the average of the previous two years. A rather short reference period was chosen for several reasons, including the proximity in time with the comparison year and the likely similar mortality patterns and age structure (especially given the absence of age-standardization). Pre-existing trends and changing demographics are, in fact, unlikely to affect forecasted estimates, as neither of these two can generally change over a short period of time, especially when examining large populations. However, in general, previous influenza pandemic peaks, or increased mortality due to harsh climatic conditions, may heavily affect forecasted values when using such a short period and only relying on the average. Thus, a careful evaluation of mortality patterns in the reference period is warranted to examine its representativity. Furthermore, the choice of not extending the comparison period beyond two years was also due to a change in the algorithm applied to identify the underlying cause of death that was implemented in the region, starting from 2018.

Average age-standardized mortality rates were from the previous 5 years is the most commonly used time window for comparison, but, despite controlling for the aging population, this approach does not take into account pre-existing time trends. Therefore, in the presence of increasing or decreasing trends, the average of rates from the previous 5 years can lead to markedly higher or lower forecasted values than those obtained through methods based on time-series analyses. 

The SARIMA and GEE models allowed us to account for seasonality and trends with high accuracy. For this reason, we chose to use data from the longest available observation period (2008–2019). The seasonal component was determined in the GEE model by using a dummy variable for each month, as opposed to the sinusoidal function (Fourier series) [13] or other approaches (wavelets or seasonal modulation) applied by other authors [27,28]. This choice likely made these algorithms more accurate in forecasting the January peaks associated with previous influenza seasonal mortality. The use of dummy variables was the best choice, given the aggregation of data on a monthly basis, but other approaches could be adopted in the presence of daily or weekly mortality data. As a sensitivity analysis, we estimated a GEE model replacing the month codified in dummy variables with a Fourier series; as a main effect, an increased smoothing was observed, associated with a lower forecasted seasonal mortality peak in January 2020 (data not shown). 

There were two main peculiarities in the observed mortality data in 2020: (1) the seasonal influenza pandemic peak of January was not present, and (2) the presence of two COVID-19 epidemic waves leading to distinct mortality peaks in months that were not usually affected by high mortality. The high mortality observed during the two COVID-19 waves was registered outside the usual period for seasonal peaks of influenza-related mortality that generally occurs in January–February. Compared to influenza mortality peaks, the pandemic was characterized by one delayed peak during the first wave (March–April 2020) and another anticipated peak during the second wave (November–December 2020). Most of the excess mortality during the pandemic waves could be explained by deaths with mention of COVID-19.

When examining all-cause mortality, characterized by a pre-existing declining trend, the average number of deaths in 2018-2019, the SARIMA and GEE estimates were similar overall and in month-by-month comparisons. The most relevant difference was observed for the average age-standardized rates of the previous 5 years. This approach did not account for the pre-existing declining trend. Thus, the magnitude of the excess observed in 2020 was underestimated, as forecasted figures were affected by the sensibly higher mortality rates registered in the previous years. The relatively low influenza-related mortality peaks in January 2018 and 2019 translated into a moderate excess mortality estimate in January 2020 with the first approach. On the other hand, forecasted figures based on the average age-standardized mortality rates in the previous 5 years were affected by very high mortality peaks in January 2015 and 2017. Hence, the observed mortality in January 2020 appeared to be significantly reduced. A similar effect likely affected the forecasted rate obtained with the SARIMA model, while only to a lesser extent when estimated with GEE. By contrast, the magnitude of excess deaths in correspondence with pandemic peaks in 2020 was similar across most models, except for the 5-year age-standardized rate, and that was always attributable to the lack of control for the pre-existing trend. 

An overall similar picture can be derived for circulatory disease mortality that represents the most common cause of death and is characterized by a pre-pandemic declining trend and a strong seasonal variation. 

Deaths with cancer as an underlying cause were not greatly affected by the pandemic. Of note, there was a slight peak in mortality in March 2020, captured by all methods, with the exception of the 5-year age-standardized mortality rates. The lack of seasonality for this cause of death translated in an estimated stable if not a slight excess death in January 2020 compared to the previous years. 

In regard to neurologic and mental disorders, due to smaller numbers, unstable estimates for raw death counts and standardized mortality rates may affect some periods characterized by low mortality (i.e., spring, summer, and autumn). However, the presence of an increasing trend has differentially affected the methods we examined. In fact, the average number of deaths and average age-standardized mortality rates approaches showed an increase in mortality, while the SARIMA and GEE models showed a stable, if not slightly decreasing, trend. Only the latter two algorithms could capture the pre-existing trends, thereby forecasting higher mortality rates for 2020, leading to a substantially lower, if not absent, excess mortality in 2020. 

The study strengths included large numbers available even for monthly mortality rates for most specific causes of death, thereby ensuring stable estimates. Furthermore, the availability of mortality data through 2008–2020 allowed us to tailor the length of the reference period based on the specific characteristics of the different methods. Longer periods were provided to methods that accounted for time trends, while shorter periods were preferred for methods that could not account for trend or those that were based on raw numbers (which could not account for neither time trends nor changes in the age structure of the study population). 

A possible limitation of this study, only for the cause-specific mortality, consisted of the change in the algorithm used to select the UCOD, starting from 2018. However, considering the broadness disease categories used in the analyses, this was unlikely to have a major impact on study results. A further limitation may have been the use of monthly as opposed to weekly or daily mortality data that may have allowed us to observe within-month fluctuations. However, for the purpose of comparing forecasting methodologies, this choice should not have hindered our results, as suggested by a previous study [29]. Lastly, all analyses were based on the UCOD. This standard approach allowed us to detect the increase in mortality from circulatory diseases during 2020. However, for other diseases where COVID-19 may have acted as a strong competing cause, the analysis of all conditions mentioned in death certificates (multiple causes of death) may have represented a better approach to assess mortality changes during the pandemic [30]. Future studies may attempt to examine cause-specific mortality as any mention on the death certificate, to avoid bias linked to the selection of the underlying cause of death with a consequent underestimation of the actual disease-specific mortality burden, an approach already adopted in several countries [30,31]. 

## 5. Conclusions

The magnitude of estimated excess mortality varied greatly based on the method used to forecast mortality figures in 2020 and was largely affected by the pre-existing trend. The results obtained with estimates based on average age-standardized mortality rates from the previous 5 years markedly diverged from all other approaches, likely due to the relatively long reference period and the impossibility to account for trends. Differences across other methods were limited. However, given the elevated flexibility of the GEE models that allowed us to include linear and non-linear trends also by means of splines, and the possibility to implement the models with numerous covariates, the GEE models might represent the most versatile option among those analyzed. Methods adopted in public health surveillance for all-cause and cause-specific mortality should take into account previous long-term trends and seasonality. However, a simple approach to surveillance (e.g., excess of absolute death counts with a short reference period) might be adopted to build a fast surveillance system, but, thereafter, results should be anyway compared with more complex regression methods.

## Figures and Tables

**Figure 1 ijerph-20-05941-f001:**
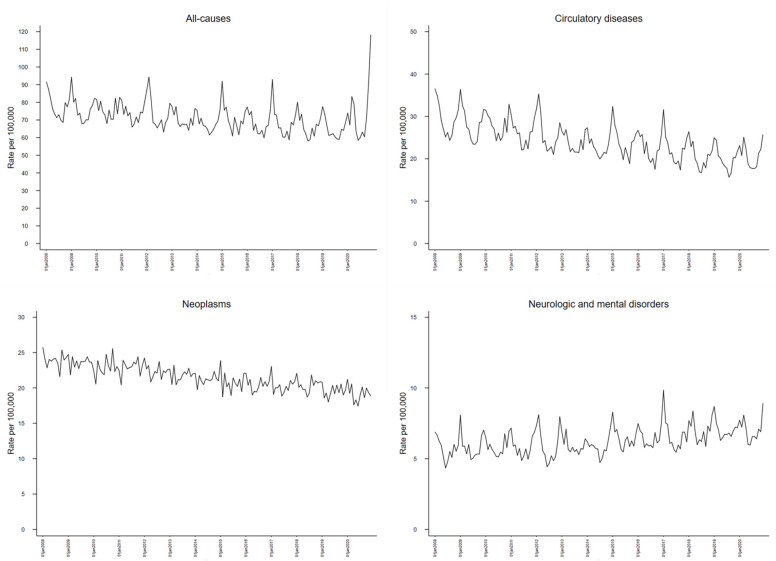
Monthly age-standardized mortality rates (per 100,000, European standard population) from all-causes, circulatory diseases, neoplasms, neurologic and mental disorders. January 2008–December 2020, Veneto Region (Italy).

**Figure 2 ijerph-20-05941-f002:**
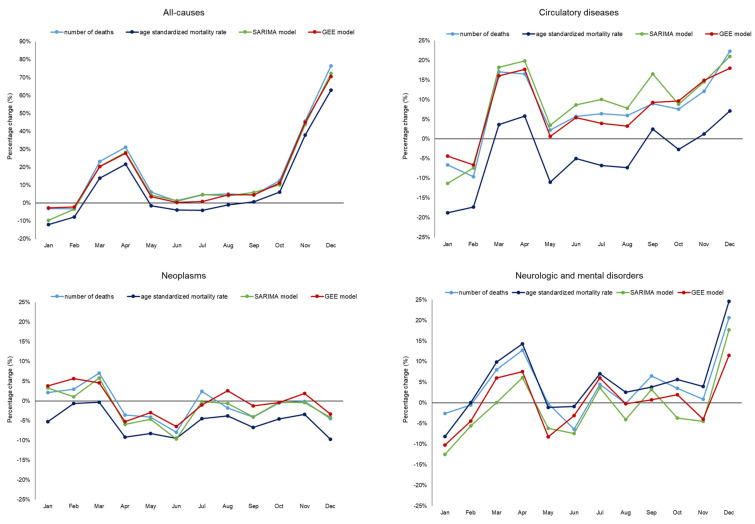
Percentage change according to different methodological approaches, for all causes and selected nosologic categories by month during 2020 (Veneto Region, Italy).

**Figure 3 ijerph-20-05941-f003:**
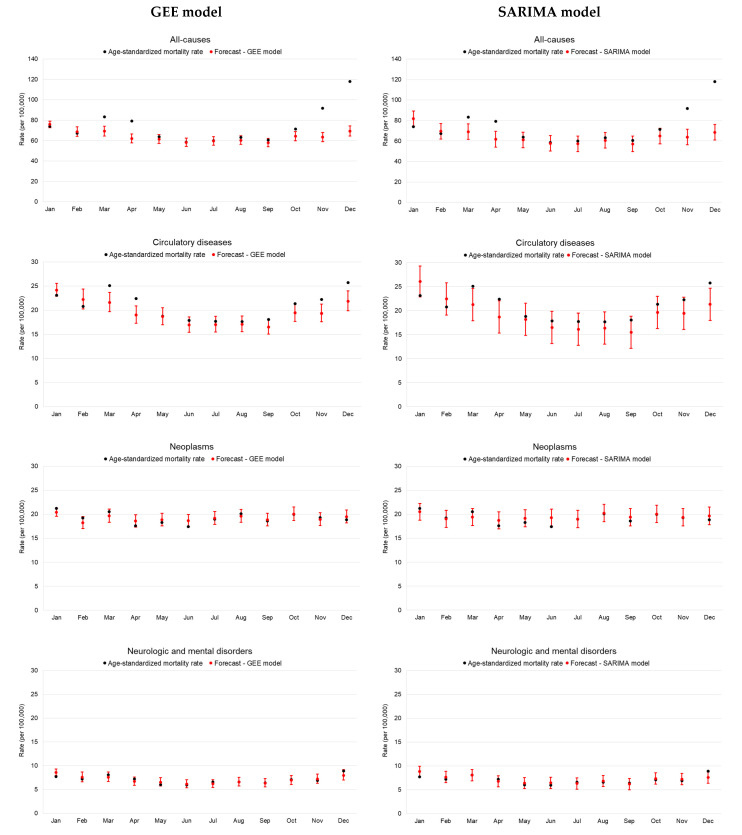
Monthly age-standardized mortality rates in 2020: observed and expected through GEE and SARIMA models for all causes and selected nosologic categories.

**Table 1 ijerph-20-05941-t001:** Number of deaths and age-standardized mortality rates registered in 2020 for all causes and selected nosologic categories, and estimated percentage changes with conditional formatting *, according to the four different methodological approaches.

	Month of Death	Deaths, n	Age-Standardized Mortality Rate	Percentage Change (%)			
				Number of Deaths ^(1)^	Age-Standardized Mortality Rate ^(2)^	SARIMA Model ^(3)^	GEE Model ^(3)^
All-causes	January	4725	73.86	−3.0%	−12.0%	−9.7%	−2.6%
	February	4280	67.20	−3.3%	−7.8%	−3.3%	−2.2%
	March	5341	83.23	23.2%	13.9%	20.5%	20.3%
	April	5098	79.11	31.2%	21.6%	28.3%	27.7%
	May	4047	63.68	6.2%	−1.5%	4.4%	3.6%
	June	3723	58.44	0.9%	−3.9%	1.3%	0.4%
	July	3831	60.07	4.6%	−4.0%	4.8%	0.8%
	August	4024	63.12	5.1%	−1.0%	4.1%	4.6%
	September	3846	60.45	4.4%	0.7%	5.9%	4.5%
	October	4570	71.41	12.5%	6.2%	10.2%	11.1%
	November	5885	91.82	45.8%	38.0%	43.9%	45.0%
	December	7603	118.10	76.5%	63.0%	72.3%	70.6%
	* **Year 2020** *	* **56,973** *	* **890.50** *	* **17.2%** *	* **9.5%** *	* **15.2%** *	* **15.7%** *
Circulatory diseases	January	1522	23.10	−6.6%	−18.7%	−11.3%	−4.4%
	February	1356	20.77	−9.6%	−17.3%	−7.3%	−6.6%
	March	1659	25.08	17.1%	3.7%	18.2%	16.0%
	April	1476	22.40	16.5%	5.8%	19.8%	17.7%
	May	1230	18.80	2.2%	−11.0%	3.5%	0.6%
	June	1175	17.90	5.7%	−5.0%	8.7%	5.5%
	July	1163	17.73	6.4%	−6.7%	10.0%	4.0%
	August	1161	17.67	6.0%	−7.3%	7.8%	3.3%
	September	1187	18.07	9.0%	2.5%	16.5%	9.3%
	October	1406	21.33	7.6%	−2.7%	8.8%	9.7%
	November	1460	22.24	12.1%	1.3%	14.5%	14.9%
	December	1689	25.74	22.3%	7.1%	21.0%	18.0%
	* **Year 2020** *	* **16,484** *	* **250.85** *	* **7.1%** *	* **−4.4%** *	* **8.4%** *	* **7.2%** *
Neoplasms	January	1294	21.20	2.1%	−5.2%	3.3%	3.8%
	February	1175	19.24	3.0%	−0.6%	1.0%	5.7%
	March	1261	20.55	7.0%	−0.3%	5.8%	4.6%
	April	1082	17.61	−3.5%	−9.2%	−5.9%	−5.3%
	May	1110	18.29	−4.1%	−8.2%	−4.6%	−2.9%
	June	1065	17.43	−7.9%	−9.4%	−9.7%	−6.5%
	July	1167	18.97	2.5%	−4.5%	−0.2%	−1.0%
	August	1237	20.14	−1.7%	−3.7%	−0.6%	2.6%
	September	1127	18.61	−4.1%	−6.7%	−4.0%	−1.2%
	October	1220	19.99	−0.5%	−4.6%	−0.3%	−0.4%
	November	1182	19.30	−0.1%	−3.4%	−0.4%	1.9%
	December	1151	18.85	−4.5%	−9.7%	−4.2%	−3.3%
	* **Year 2020** *	* **14,071** *	* **230.19** *	* **−1.0%** *	* **−5.5%** *	* **−1.6%** *	* **−0.1%** *
Neurologic and mental disorders	January	509	7.71	−2.6%	−8.1%	−12.5%	−10.2%
	February	470	7.23	−0.5%	0.1%	−5.6%	−4.4%
	March	529	8.08	8.1%	9.9%	0.1%	6.1%
	April	472	7.20	12.8%	14.3%	6.1%	7.6%
	May	394	6.00	−0.1%	−1.1%	−6.2%	−8.2%
	June	386	5.96	−6.4%	−0.9%	−7.4%	−3.1%
	July	430	6.56	4.5%	7.1%	3.7%	6.0%
	August	435	6.57	−0.1%	2.6%	−4.0%	−0.2%
	September	422	6.40	6.6%	3.8%	3.3%	0.8%
	October	466	7.08	3.6%	5.7%	−3.7%	2.0%
	November	451	6.91	0.9%	4.0%	−4.5%	−4.0%
	December	585	8.90	20.6%	24.6%	17.7%	11.5%
	* **Year 2020** *	* **5549** *	* **84.60** *	* **4.0%** *	* **5.1%** *	* **−1.3%** *	* **0.3%** *

* Conditional formatting is specific for monthly and yearly variation, and per cause, with a color scale where green defines the lowest variation and red the highest. ^(1)^ reference value: average deaths 2018–2019. ^(2)^ reference value: average age-standardized mortality rate 2015–2019. ^(3)^ reference value: forecast age-standardized mortality rate 2020 using historical mortality trends 2008–2019.

**Table 2 ijerph-20-05941-t002:** Generalized estimating equation model for the monthly age-standardized mortality rate by all-causes and selected nosologic categories.

	All-Causes		Circulatory Diseases		Neoplasms		Neurologic and Mental Disorders	
	Estimate	(95% CI)	Estimate	(95% CI)	Estimate	(95% CI)	Estimate	(95% CI)
Time trend–year	0.98	(0.98–0.99)	0.97	(0.96–0.97)	0.98	(0.98–0.98)	1.02	(1.02–1.02)
January	1.00	#	1.00	#	1.00	#	1.00	#
February	0.91	(0.88–0.93)	0.92	(0.89–0.96)	0.89	(0.87–0.92)	0.88	(0.83–0.93)
March	0.91	(0.89–0.94)	0.90	(0.87–0.93)	0.97	(0.94–0.99)	0.88	(0.84–0.93)
April	0.82	(0.80–0.84)	0.79	(0.77–0.82)	0.91	(0.89–0.94)	0.77	(0.73–0.82)
May	0.81	(0.79–0.84)	0.78	(0.75–0.81)	0.93	(0.90–0.95)	0.76	(0.72–0.80)
June	0.77	(0.75–0.79)	0.71	(0.69–0.74)	0.92	(0.90–0.95)	0.71	(0.67–0.75)
July	0.79	(0.77–0.81)	0.72	(0.69–0.74)	0.95	(0.92–0.97)	0.71	(0.68–0.75)
August	0.80	(0.78–0.83)	0.72	(0.70–0.75)	0.97	(0.95–1.00)	0.76	(0.72–0.80)
September	0.77	(0.75–0.79)	0.70	(0.67–0.73)	0.93	(0.91–0.96)	0.73	(0.69–0.77)
October	0.86	(0.83–0.88)	0.83	(0.80–0.86)	1.00	(0.97–1.02)	0.80	(0.76–0.84)
November	0.85	(0.82–0.87)	0.82	(0.79–0.86)	0.94	(0.92–0.97)	0.82	(0.78–0.87)
December	0.93	(0.90–0.95)	0.93	(0.90–0.97)	0.97	(0.94–1.00)	0.91	(0.87–0.96)

# Reference category.

## Data Availability

The data presented in this study are available on request from the corresponding author. The data are not publicly available.

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
