# Peer review of "Estimating Overall and Cause-Specific Excess Mortality during the COVID-19 Pandemic: Methodological Approaches Compared"

_ijerph, 2023, doi:10.3390/ijerph20115941_

Round 1

Reviewer 1 Report

The sample size is not enough and the results cover only a region!

Author Response

We thank the Reviewer for the positive evaluation of the manuscript, however, in regards to the low sample size, the population is still relatively large (4.9 million) and we believe numbers were sufficient to obtain stable estimates for most of the analyses, and that this has not hindered between-methods comparison. 

Reviewer 2 Report

Line 52- Can you give examples of what different approaches were handled with respect to mortality and why they were not preferable.

Line 61- Explain more rational for selecting the Veneto region.  How this region is similar or distinct from other regions.  Elaborate on aspects of demography from this region those would affect public health, such as socio-economic status, education, healthcare system, access, reliability of public health in the region.

Line 162- Fig 2, Graphs of Neurology and circulatory and Neurologic disorders cannot be seen completely.  Can you re-arrange borders/ margins of page such that the figure would be visible in complete manner.

Line 278- 280- Kindly explain why seasonal January peak of influenza was not seen in 2021 and rational for not having high mortality the two Covid-19 waves.

Line 314- Would you elaborate on how strengths and conclusions from this study would help to improve public health at local level for the region and at global level.

Author Response

We thank the Reviewer for the comments and we report here below the point by point answers:

Comment 1. Can you give examples of what different approaches were handled with respect to mortality and why they were not preferable.

Answer. Common approaches adopted to assess excess mortality during the pandemic include the four methods compared in the present paper. This was reported only in the first paragraph of the Discussion of the original manuscript, and is now briefly summarized at the end of the Introduction section of the revised paper.

Comment 2. Explain more rational for selecting the Veneto region. How this region is similar or distinct from other regions. Elaborate on aspects of demography from this region those would affect public health, such as socio-economic status, education, healthcare system, access, reliability of public health in the region.

Answer. Main characteristics of the Veneto region are now presented in the Introduction section: there is a regional health system funded by general taxation with free access at the point of service, at least for emergency and hospital care. The region was among the first and most severely hit areas by the pandemic at the beginning of 2020, and this led to a previously documented strong public health response [Russo 2020].

Comment 3. Fig 2, Graphs of Neurology and circulatory and Neurologic disorders cannot be seen completely. Can you rearrange borders/ margins of page such that the figure would be visible in complete manner.

Answer. We’re sorry for the error in the pdf production, that has been fixed in the revised manuscript.

Comment 4. Kindly explain why seasonal January peak of influenza was not seen in 2021 and rational for not having high mortality the two Covid-19 waves.

Answer. According to syndromic surveillance carried out in Italy since several years, influenza activity in January 2020 was low. It is now more clearly reported that the high mortality was observed during the two COVID-19 waves, and this excess was registered outside the usual period for seasonal peaks of influenza-related mortality, generally observed in January-February (later for the first pandemic wave, March-April 2020; sooner for the second wave, November-December 2020).

Comment 5. Would you elaborate on how strengths and conclusions from this study would help to improve public health at local level for the region and at global level.

Answer. It is now reported in the conclusions that methods adopted in public health surveillance of general and cause-specific mortality should take into account previous long-term trends and seasonality of mortality by cause. A simple approach to surveillance (e.g. excess of absolute deaths counts with a short reference period) might be adopted to build a fast surveillance system, but thereafter results should be anyway compared with more complex regression methods.

Reviewer 3 Report

This manuscript compares four methods in their estimates of excess mortality during 2020 in the Veneto region in Italy, to get an estimate of COVID-19 related mortality.

I thought the manuscript was well written and clear, however, the final conclusion or recommendation resulting from the analyses was not clear to me. I have several comments, questions and suggestions.

1. Table 1 is very long and not clearly formatted. Some formatting by highlighting diverging estimates could help in inspecting the numbers in this table. Alternatively, they could be presented in an appendix, as they convey the same information as Figure 2.

2. Figure 2 is not well placed in the manuscript: I could not inspect half of the figure.

3. Only the GEE model was produced with uncertainty intervals. I assume the other models can also be produced with uncertainty intervals (in particular the SARIMA model). What is the reason only point estimates are compared? I think it would add great value to learn about the differences in the precision of the different types of estimates.

4. It is unclear to me why these four models were chosen for the comparison of the excess mortality.

5. I am curious as to why only the excess mortality was modeled, and no attempt to validate these results with the reported COVID-19 related mortality was made. While the reported deaths are certainly an underestimate, it would be an interesting comparison to anchor the estimates to.

6. The authors are advising the GEE from the fact that it is the most versatile model. However, I did not see a consideration for the potential sources of bias that could be introduced by considering a more complex model, nor the ease of use of such a model as compared to the simpler models -- in what contexts do the authors think which model should be used?

7. While indeed four methods used to estimate excess mortality are compared, it is by no means a fair comparison, as different pieces of information are included in the estimates. Naturally, as more information gets included, estimates are likely to get more precise. I missed a validation of the estimates, or another source of describing the potential bias from each method, through, for example, a small simulation study.

Author Response

We thank the Reviewer for the comments and the positive consideration given to our manuscript, and we attach below the point by point answers.

Comment 1. Table 1 is very long and not clearly formatted. Some formatting by highlighting diverging estimates could help in inspecting the numbers in this table. Alternatively, they could be presented in an appendix, as they convey the same information as Figure 2.

Answer. Table 1 formatting has been changed in the revised manuscript to make the presentation of the main results clearer.

Comment 2. Figure 2 is not well placed in the manuscript: I could not inspect half of the figure.

Answer. We’re sorry for the error in the pdf production, that has been fixed in the revised manuscript.

Comment 3. Only the GEE model was produced with uncertainty intervals. I assume the other models can also be produced with uncertainty intervals (in particular the SARIMA model). What is the reason only point estimates are compared? I think it would add great value to learn about the differences in the precision of the different types of estimates.

Answer. Figure 3 has been modified by adding uncertainty intervals for the SARIMA model, also to  allow for direct visual comparison of the estimate precision of the two regression approaches adopted.

Comment 4. It is unclear to me why these four models were chosen for the comparison of the excess mortality.

Answer. The four methods compared are among the most common approaches adopted in the literature to assess the impact of the pandemic; this has been added at the end of the Introduction of the revised manuscript.

Comment 5. I am curious as to why only the excess mortality was modeled, and no attempt to validate these results with the reported COVID-19 related mortality was made. While the reported deaths are certainly an underestimate, it would be an interesting comparison to anchor the estimates to.

Answer. In the revised manuscript, Supplementary Figure 1 has been added comparing excess overall deaths (approach 1 based on death counts) with deaths attributed to/with mention of COVID-19. However, it must be acknowledged that this comparison add only to the evaluation of overall (and not cause-specific) mortality, and can be tricky due to undiagnosed COVID-19, especially in the first epidemic wave when diagnostic capabilities were limited, and to rules set by the WHO for attribution of deaths to COVID-19, with a possible underestimation of other causes of death during the pandemic. The above have been added to the Results and Discussion section of the revised paper.

Comment 6. The authors are advising the GEE from the fact that it is the most versatile model. However, I did not see a consideration for the potential sources of bias that could be introduced by considering a more complex model, nor the ease of use of such a model as compared to the simpler models -- in what contexts do the authors think which model should be used?

Answer. We’ve expanded conclusions of the revised paper: methods adopted for surveillance of general and cause-specific mortality should take into account both long-term trends and the seasonal pattern of mortality. Simpler approaches to surveillance (e.g. excess of deaths counts with respect to a short reference period) might be useful to build-up a fast surveillance system, but thereafter results should be anyway compared with more complex regression methods.

Comment 7. While indeed four methods used to estimate excess mortality are compared, it is by no means a fair comparison, as different pieces of information are included in the estimates. Naturally, as more information gets included, estimates are likely to get more precise. I missed a validation of the estimates, or another source of describing the potential bias from each method, through, for example, a small simulation study.

Answer. As regards overall mortality, some clues can be added by comparison with deaths attributed to COVID-19, considering all limits of these latter approach (see also Answer to Comment 5).  For cause-specific mortality, we think that there is no simple validation of estimates; some additional information can be obtained by comparing results obtained by the multiple causes of death approach, already adopted both in our region and in other countries [Fedeli 2022, Spijker 2023]. This has been expanded at the end of Discussion of the revised paper, addressing also the Academic Editor’s suggestion.